



# Present and future trends of extreme short-term rainfall events in Germany, by downscaling convective environments of ERA5 and a CMIP6 ensemble

Gerd Bürger[1] and Maik Heistermann[1]

[1]Universität Potsdam, Karl-Liebknechtstr. 24-25, 14476 Potsdam, Germany

**Correspondence:** Gerd Bürger (gerd.buerger@uni-potsdam.de)

**Abstract.** For the four main quadrant regions of Germany we study the possibility of projecting the occurrence of extreme convective rainfall events, as monitored by the CatRaRE database, into the future, based on CMIP6 projections of corresponding convective environments. We characterize such environments by using the atmospheric profile derivates convective available potential energy (*cape*) and convective inhibition (*cin*), along with model-simulated convective precipitation (*cp*). The convective environments are linked to the small-scale CatRaRE events by classifying the corresponding ERA5 fields according to the concurrent occurrence of such events. Classifiers are conventional machine-learning procedures along with more modern deep learning schemes. Positive centennial trends are identified for both antagonists *cape* and *cin*, serving as a source for large uncertainties of the corresponding CatRaRE-type trends. Their full distribution is analyzed using ANOVA, based on the factors of event severity, greenhouse gas emissions, climate models, classifiers, and region. Beyond all uncertainty, positive trends outweigh the negative ones for all regions.

## 1 Introduction

Flash-floods are among the most destructive environmental hazards. The areas mostly affected are mountainous regions as well as urban agglomerations. As a result, flash-floods are for Germany the most costly type of extreme events, with an estimated damage of more than 70B EUR for the last two decades (Trenczek et al., 2022).

It is known that a warmer atmosphere contains more water vapor (the so-called Clausius-Clapeyron (CC) law), giving rise to enhanced downpour once a heavy precipitation event is under way (CC scaling, cf. Fowler et al., 2021). There is also evidence of this happening more often, even more so for the strongest events (Myhre et al., 2019). This pertains in particular to short-term heavy convective summerly rainfall that is typically the cause of flash floods. To quantify future impacts, state-of-the-art climate models increasingly implement the very complex physics of convection into their thermodynamic routines (Fosser et al., 2020; Kendon et al., 2021; Lin et al., 2022; Lucas-Picher et al., 2021). These being computationally expensive, corresponding simulations remain limited and statistical estimations uncertain. To overcome such limitations, convection is sometimes emulated in climate models, using a whole spectrum of schemes ranging from simple parametric approaches to full-blown emulators driven by artificial intelligence (AI) (Gentine et al., 2018; O'Gorman and Dwyer, 2018); see also Ukkonen and Mäkelä (2019). More conventionally, empirical post-processing schemes known as 'statistical downscaling' are employed.





This is facilitated by using, for example, variables that describe the atmospheric conditions prone to convection – so called
*convective environments* – and hence to heavy precipitation. This way, century-long projections of short-term regional or local
future rainfall and correspondingly more stable statistics of rainfall extremes can be obtained (Bürger et al., 2019; Bürger and
Heistermann, 2023), henceforth BH). Still, many questions remain unsettled regarding the interplay between (present or future)
atmospheric convective environments and heavy local precipitation (Meyer et al., 2022).

For Germany, an inventory of extreme rainfall events has recently been compiled from more than 20 years of weather radar
records (CatRaRE, cf. (Lengfeld et al., 2021)). In a first effort to project CatRaRE-type events into the future, BH applied a
classification approach to downscale a single future simulation of the EURO-CORDEX suite. As classifiers, conventional sta-
tistical downscaling schemes were benchmarked against more modern AI-based models. Regardless of method, CatRaRE-type
extreme events were projected to increase significantly throughout the century. This study is now extended in various direc-
35 tions: while BH had used a binary predictand (event/no event), this is now extended to include multivariate regional predictands
as well as different event magnitudes. And most notably, instead of using a single model of the older CMIP5/CORDEX suite,
here we use the new CMIP6 models directly, and we do so with as many of them as possible to obtain thorough uncertainty
estimates of our results.

## 2    Methods and data

### 2.1    CatRaRE

The basis of CatRaRE is a 22-year long record of hourly precipitation fields of 1 km resolution over Germany, which itself is
derived from weather radar observations (Winterrath et al., 2018). From these precipitation fields, CatRaRE identifies spatio-
temporally contiguous objects (events) in which rainfall exceeds warning level 3 (as defined by the Germany Meteorological
Service). This threshold corresponds to 25 mm in one hour or 35 mm in six hours. Each event is assigned a location and
45 timestamp, using its centroid and mean of start and end date, respectively, as well as a set of additional metrics. One of these
metrics is the extremeness index $E_{T,A}$ of Müller and Kaspar (2014), which combines area, duration and intensity measures.
Focusing on convection, we select events that took place in the summer months May to August with a duration of at most 9
hours, leaving a total of 20824 cases. As a first glimpse, Fig. 1 displays our selection of CatRaRE events grouped into single
years, along with the corresponding annual 99 %-, 99.9 %-, and 99.99 %-percentiles of $E_{T,A}$ and their trends. While the 99
50 %-percentiles show no trend (p>0.5), the extreme (99.9 %) and very extreme (99.99 %, equaling the maximum) percentiles
reveal, with growing significance (p<0.02 and p<0.002, resp.), clear upward trends. Note that although we are dealing with
extreme precipitation, the integrating effect of the extremeness measure $E_{T,A}$ leads to quantities whose trend residuals appear
normally distributed and the trends themselves quite robust.

To further study these very extreme cases, we select events that have $E_{T,A} \geq 8.7$ and $E_{T,A} \geq 20.7$, corresponding to the
55 overall 90 % and 99 % percentile and representing the 2040 and 208 strongest cases, respectively. The severity classes, denoted
as $P_{00}$, $P_{90}$, and $P_{99}$, respectively, should be sufficiently large to undergo the statistical modeling as described below.





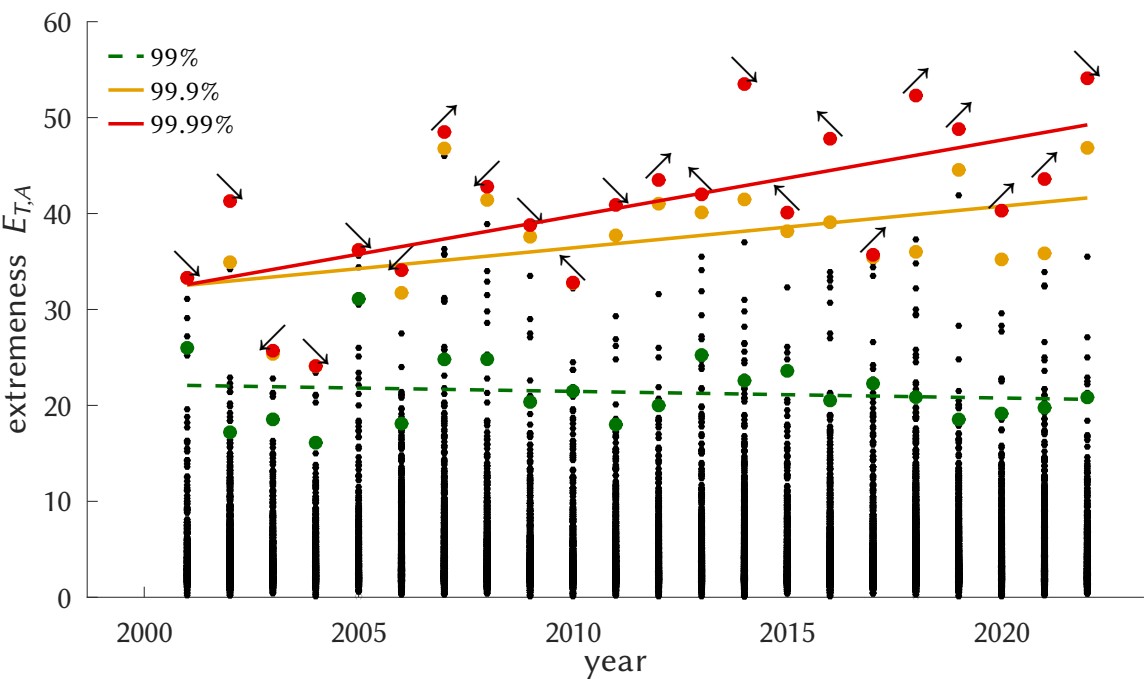

**Figure 1.** For each year, all relevant CatRaRE events ($\leq 9h$, May–Aug) are shown (black dots), along with their 99 %-, 99.9 %-, and 99.99 %-percentiles and corresponding regression trendlines (colored dots and lines) and p-values $p \approx 0.6$, $p \approx 0.02$, and $p \approx 0.002$. The arrows attached to the red dots point to the region quadrant of the respective event.

## 2.2 Atmospheric data

We use observed (reanalyzed) atmospheric fields from ERA5 for the summer months of the years 1979–2022 (Hersbach et al., 2020) and corresponding fields as simulated for present (1960–2014) and future (2015–2100) by the current version of the

Coupled Model Intercomparison Project (CMIP6), available on the Earth System Grid (https://pcmdi.llnl.gov/CMIP6). As convective environments (predictor fields) we use convective available potential energy (*cape*), convective inhibition (*cin*), and (model-simulated) convective rainfall (*cp*). As the name suggests, *cape* is a measure of the atmosphere's potential to freely develop convection between the level of free convection (LFC) and the equilibrium level (EL) where buoyancy stops. *cin*, on the contrary, measures the stability below: starting at the surface level (SFC), it quantifies the effort that is needed to overcome

LFC and hence initiate convection; a typical force for such upward movement are orographically forced or frontal updrafts. It follows that the local effect of *cape* is mostly on rainfall intensity while for *cin* it is on rainfall occurrence (or frequency). Both, *cape* and *cin*, are integral measures of the temperature and humidity profiles, basically measuring each level's contribution





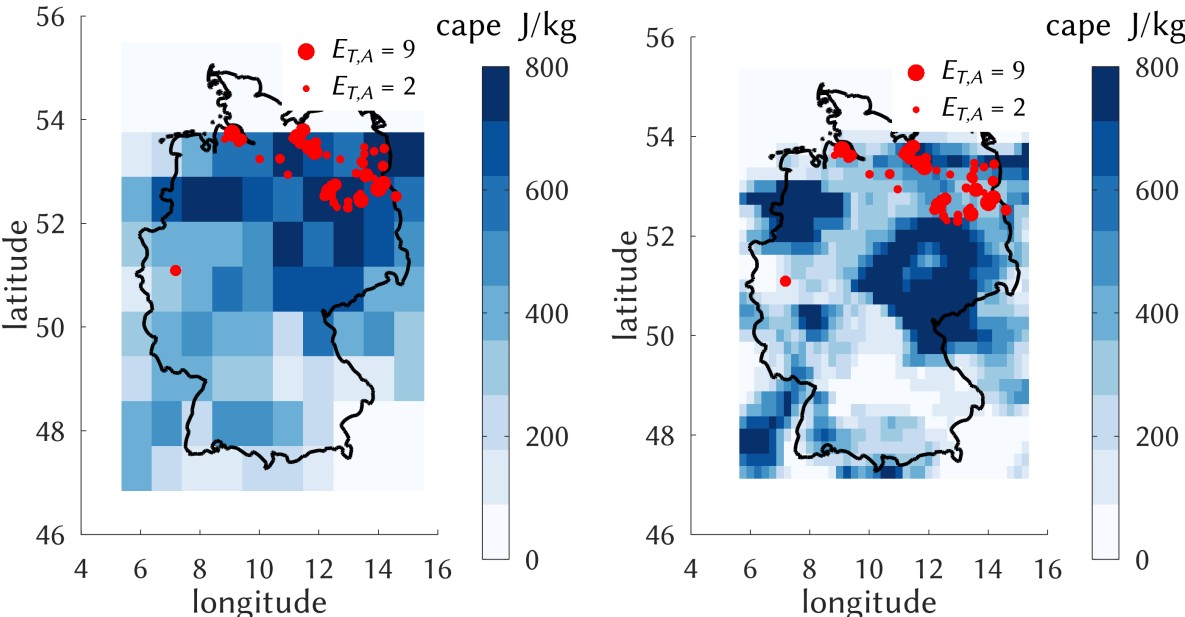

**Figure 2.** Left: Typical strong convective constellation on the afternoon of June 17, 2001, with a corresponding *cape* field at resolution 0.8×0.8, as calculated from ERA5 profiles using Eq. 1 (see text) and classified as a Northeast (NE) event; red dots indicate CatRaRE events and their extremeness $E_{T,A}$, with $2 \leq E_{T,A} \leq 9$. Right: The same situation with the original ERA5 resolution of 0.25×0.25 as used by BH.

to the parcel's buoyancy. The latter is essentially given by the relative difference of the parcel's virtual temperature and the temperature of environment, $T_{v,p}$ and $T_{v,e}$, respectively, so that *cape* and *cin* are obtained from Eqs. 1 and 2:

$$\text{cape} = \int_{z_{\text{LFC}}}^{z_{\text{EL}}} g \left( \frac{T_{v,p} - T_{v,e}}{T_{v,e}} \right) dz \tag{1}$$

$$\text{cin} = \int_{z_{\text{SFC}}}^{z_{\text{LFC}}} g \left( \frac{T_{v,p} - T_{v,e}}{T_{v,e}} \right) dz \tag{2}$$

As *cape* and *cin* are not provided by the CMIP6 models they had to be calculated from these profiles, following Bolton (1980). ERA5 provides convective environments, but for consistency, Eqs. 1 and 2 were also used for ERA5. Note, however, that upper-air measurements from observations are sparse, so that atmospheric profiles, from reanalysis or simulations, remain uncertain (Taszarek et al., 2021). Convective precipitation (*cp*) itself is provided by all models (Emmenegger et al., 2024).

To capture uncertainty, we have attempted to obtain as many of the CMIP6 simulations as possible. Several limitations applied: because we use an anomaly approach (see below), respective reference (historical) simulations of the same model are required; furthermore, only models were used whose temporal resolution of the vertical profiles was at least 6-hourly, and *cp*





was interpolated or aggregated to the same resolution. Similarly, we transformed all fields to a unique 10×10 grid between the corners [5.75E 47.25N] and [15.25E 55.25N], by interpolation or aggregation, and for the former we allowed only models with a maximum original resolution of 1.5×1.5 degrees. A summary of all simulations is shown in Table 1. For any given GCM and field variable, the reference climate was calculated as mean and standard deviation across all grid points and historical realizations of the same model for the reference period 1981–2010; from that, standardized anomalies are formed to enter the classification.

**Table 1.** Summary of CMIP6 simulations with realizations r1, r2, r3

| GCM | historical (HIST) | SSP126 | SSP585 |
|---|---|---|---|
| CMCC-CM2-SR5 | r1 | r1 | r1 |
| CMCC-ESM2 | r1 | r1 | |
| CNRM-CM6-1 | r1, r2, r3 | r1 | r1 |
| CNRM-ESM2-1 | r1 | r1 | |
| MPI-ESM1-2-HR | r1, r2 | r1, r2 | r2 |
| NorESM2-MM | r1, r3 | | r1 |

A typical constellation of ERA5 *cape* and the concurrent state of CatRaRE from June 17, 2001 is shown in Fig. 2. It was a day with strong atmospheric convectivity (*cape* > 700 J/kg) that led to numerous heavy thunderstorms over Northeast Germany, including the cities of Hamburg and Berlin.

## 2.3 Classification

In order to investigate whether any change in the frequency of heavy rainfall varies regionally within Germany, we subdivide the area of Germany into four quadrants (NW, NE, SW, SE), defined by maximum and minimum longitude and latitude coordinates and respective centers. Any 6-hour interval $d$ is now probabilistically classified as belonging to

– 0, if no event exists whose timestamp falls into $d$;

– R, if more than half of the events that fall into $d$ lie within the region R, with R ∈ {NW, NE, SW, SE};

– 1, otherwise.

As an example, the 6h-interval after June 17, 2001, 12:00, pertaining to Fig. 2 is classified a NE extreme event. Note that no further distinction is made, for example, with regard to multiple, almost simultaneous events in a single quadrant. Note also that the classification depends on the particular shape of the German border although its small-scale details (an event might occur just outside Germany) have certainly no representation in the atmospheric predictor fields. This is the same kind of limitation that station downscaling shows in general, and cannot be avoided. The overall statistics are summarized in Table 2.

The focus of this study lies in estimating the evolution of these class rates through time, and specifically whether the rates are affected by climate change.



**Table 2.** Case distribution (in % of all events) across regions under different percentiles

| cases | 0 | NW | NE | SW | SE | 1 |
|---|---|---|---|---|---|---|
| $P_{00}$ | 73.7 | 3.1 | 2.8 | 9.5 | 5.5 | 5.4 |
| $P_{90}$ | 91.2 | 1.5 | 1.3 | 3.0 | 1.6 | 1.4 |
| $P_{99}$ | 98.4 | 0.4 | 0.3 | 0.5 | 0.3 | 0.1 |

## 2.4 Conventional ("Shallow") and Deep Learning models

As classifiers, we employ two groups of empirical schemes, one using more conventional statistical or machine learning tools, and another representing deep learning (DL) neural networks; they are described in more detail in BH and summarized in Table 3.

**Table 3.** The conventional ("shallow") and deep learning classifiers.

| Name | Notes/resolution | layers | # Params ($\times 10^3$) | Source |
|---|---|---|---|---|
| **Conventional models** | | | | |
| Lasso logistic regression (LASSO) | Cross-validated penalty (28 predictors) | | | McIlhagga (2016) |
| Random forests (TREE) | 200 trees | | | Jekabsons (2016) |
| Shallow neural network (NNET) | 2 hidden layers with 7 and 3 neurons | | | Octave |
| Logistic regression (NLS) | Nonlinear least squares | | | Octave |
| **Deep learning models** | | | | |
| Logreg (2023) | 32×32 | 1 | 6 | BH |
| CIFAR-10 (2014) | 32×32 | 4 | 80 | Krizhevsky et al. (2017) |
| Simple (2023) | 32×32 | 3 | 300 | BH |
| ResNet (2016) | 32×32 | 22 | 300 | He et al. (2016) |
| LeNet-5 (1989) | 28×28 | 4 | 400 | LeCun et al. (1989) |
| DenseNet (2016) | 32×32 | 159 | 1000 | Huang et al. (2017) |
| ALL-CNN (2014) | 32×32 | 9 | 1000 | Springenberg et al. (2014) |
| GoogLeNet (2014) | 224×224 | 76 | 10000 | Szegedy et al. (2015) |
| AlexNet (2012) | 227×227 | 8 | 60000 | Krizhevsky et al. (2017) |

## 2.5 Exceedance counting

Prolonging the trends of Fig. 1 into future decades would represent a quite simplistic, zero-order projection of $E_{T,A}$, concluding that, for example, the 99.99 % percentiles grow from a starting level near $E_{T,A} = 30$ to one of $E_{T,A} = 50$ merely 20 years later, which amounts to a centennial trend of about 100 $E_{T,A}$ units. This approach is appealing but certainly not much more than heuristic. Our probabilistic classification scheme described above, instead, employs the full information contained in





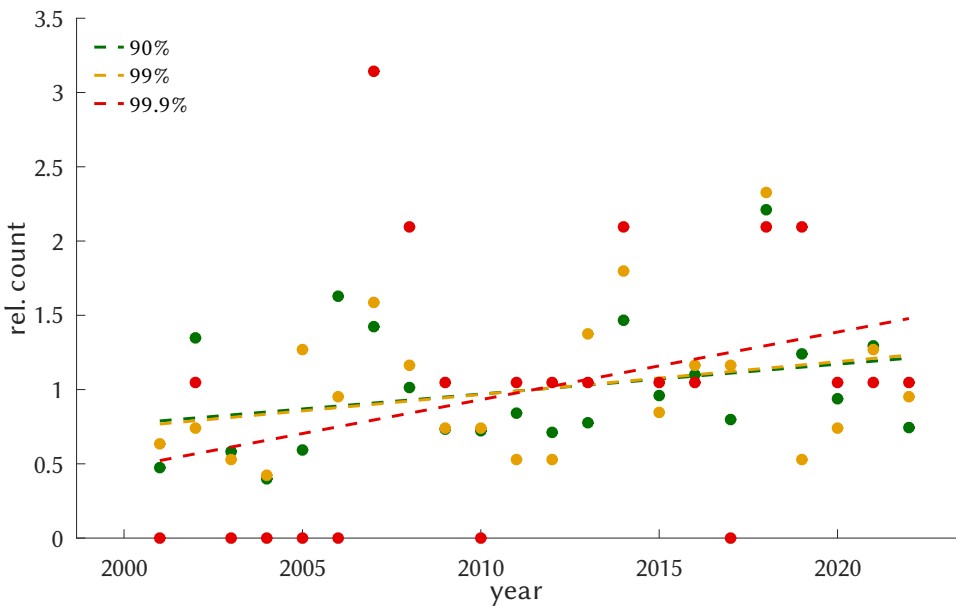

**Figure 3.** For each percentile P (90 %, 99 %, 99.9 %), a single dot represents the annual exceedance count relative to the long-term average. The corresponding trends are positive but insignificant, indicated by the dashed linestyle

present and future convective atmospheric environments on a sub-daily basis. To relate class probabilities to Fig. 1, we show in Fig. 3 for the percentiles $P_{90}$, $P_{99}$, and $P_{99.9}$ the corresponding annual relative exceedance counts normalized by the overall exceedance count. The respective trends are all positive, however non-significant, with stronger trends for the more extreme cases, similar to Fig. 1.

One cannot get around the sample size limitation, however: as mentioned, only 208 cases remain when the analysis is limited to $P_{99}$ and above (applied to all cases instead of annually), and that number shrinks to 21 for $P_{99.9}$ (and to 2 for $P_{99.99}$).

## 3    Results and discussion

### 3.1    Training and testing the classifiers with ERA5

The CatRaRE version used in this study covers the period from 2001 to 2022. By limiting the analysis to the summer months
from May to August, a total of 10824 6-hourly ERA5 fields remains to be classified according to the local conditions of CatRaRE. For the technical details of the probabilistic predictions, we refer to BH. The cases are split into a calibration (train) and validation (test) period of 2001–2011 and 2012–2022, respectively. For the DL training, cross-entropy is used as a loss function, and the Ranked Probability Skill Score (RPSS) is used to assess the classification performance (RPSS is the skill score corresponding to the relative deviance of the predicted probabilities from the actual outcome).





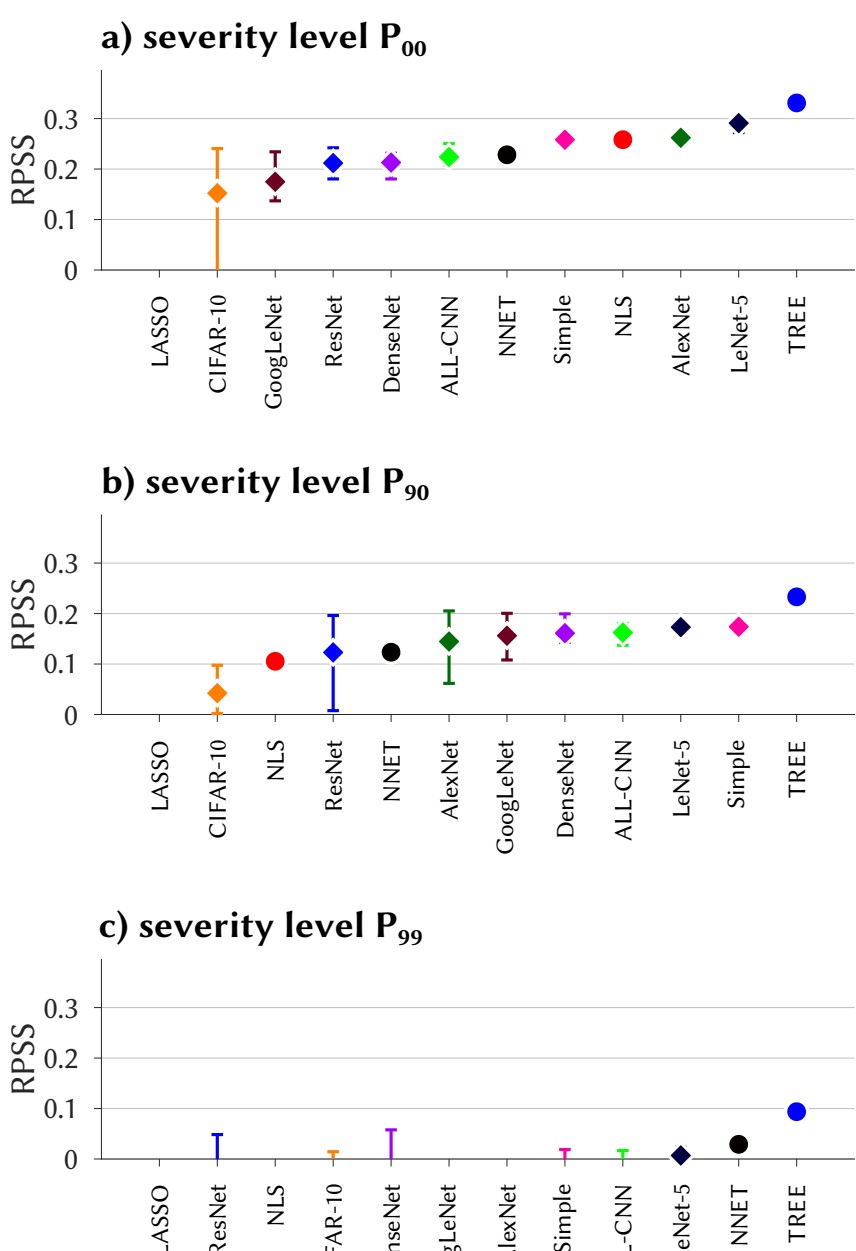

**Figure 4.** Ranked probability skill score for the 12 classifiers, based on all CatRaRE events (a, $P_{00}$) and on the upper extremes $P_{90}$ and $P_{99}$ (b, c). For visibility, only positive values are depicted.



Fig. 4 shows the scores for the respective severity classes $P_{00}$, $P_{90}$, and $P_{99}$. For all, the TREE classifier outperforms all others markedly, including the DL methods, with scores of RPSS > 0.3 for $P_{00}$ and RPSS > 0.2 for $P_{90}$, in accordance to the findings of BH; but note that for $P_{99}$, there is hardly any skill left. For $P_{90}$, some of the DL methods (ResNet, AlexNet, GoogLeNet) show a fairly large RPSS-range with maximum scores near that of TREE, but it is unclear how the stochastic nature of their training algorithm (Brownlee, 2018, cf.) can be 'tricked' into finding that maximum every time. Compared to such highly sophisticated DL frameworks, the Simple DL with just one convolutional and one dense layer performs remarkably well, and that with almost no spread, even ranking second after TREE for the $P_{90}$ case.

## 3.2 ERA5 and CMIP6 driving the trained classifiers

The trained classifiers are now applied to the observed (reanalyzed) and simulated atmospheric fields. The results are predicted probabilities for each of the 6 classes, cf. 2.3, at 6-hour intervals from the respective summer months.

Our analysis starts by a case study in which we check whether the June 2001 event (see Fig. 2) is captured by our classification of the ERA5 fields. Fig. 5 shows the probability forecasts from the TREE classifier for the cases $P_{00}$ and $P_{90}$. For both, a clear tendency is seen to generate elevated probabilities for non-zero classes towards mid-month. For $P_{00}$, the SE and NW classes dominate at the start, the NE region is persistently prominent at mid-month, and the South again dominates in the decay phase. For $P_{90}$, the evolution is much clearer: throughout the month of June there is essentially zero activity, to be interrupted only by three strong 6-hour periods at mid-month, one occurring in NW and the other two in NE.

To allow for a view at longer time scales, we form annual (i. e. May–Aug) averages of the resulting 6-hourly probabilities as obtained from the classifiers, and from now on focus entirely on these annual values and their centennial trends. We start by showing in Fig. 6 the results for the TREE classifier for class 1 and the entire ERA5 period 1979–2022; the other classifiers and regions are similar, cf. Fig.S1–S4. The simulations quite well reproduce the observed base rate of class 1 for all severities $P_{00}$, $P_{90}$, and $P_{99}$ of CatRaRE (cf. Table 2); interannual variability appears somewhat reduced. The ERA5-based trends are significantly positive throughout, with values of 3.1, 1.1, and 0.1 %pt per century for the three severities, respectively. The simulated annual values are strongly correlated with observations for $P_{00}$ and $P_{90}$ ($\rho$=0.67 and $\rho$=0.43 for the validation period, resp.); such levels of correlation are somewhat surprising given the relatively poor sub-daily performance (cf. Figure 4), but it adds confidence to using annual trends as the main target quantity. For $P_{99}$, the performance is not easy to judge since, for example, observations have virtually no variability after 2015.

Now we turn to the simulated atmospheres as provided by the CMIP6 ensemble of climate simulations of Table 1. Given the 20 entries of the table, the 3 severity classes, the 6 regions, and the 12 classifiers, we have a total of $20 \times 3 \times 6 \times 12$ = 4320 centennial trends. For the illustration of individual results we confine ourselves to a few typical outcomes here and several more in the Supplementary Information (SI). The trend distribution is analyzed by discarding zero-skill classifiers and the two non-regional classes 0 and 1, leaving a total of 2400 regional trends. In 3.3 that distribution is subject to an ANOVA to disentangle the relevant sources of uncertainty. We start by showing in Fig. 7, similar to Fig. 6 for ERA5, examples for the SW region from two different GCMs. The TREE classifier is applied here to the climate scenarios HIST, SSP126, and SSP585 as simulated by CMCC-CM2-SR5 and by MPI-ESM1-2-HR. For the former, the simulations appear consistent with the overall





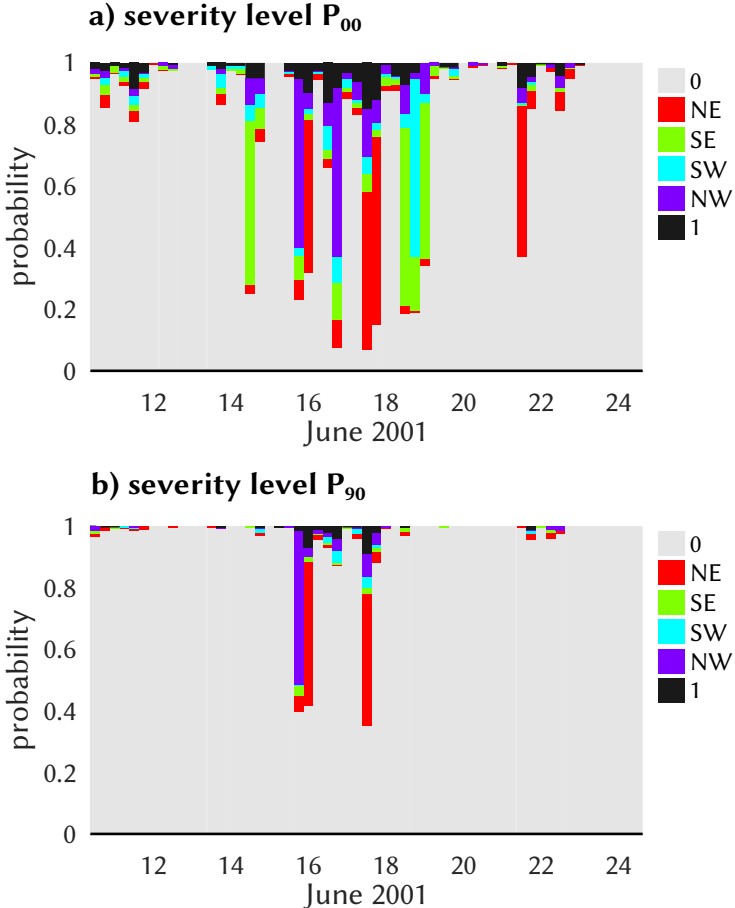

**Figure 5.** 6-hourly probabilistic predictions of regional classes around the June 17, 2001 events, as obtained by the TREE classifier from ERA5 convective environments for the two severity levels $P_{00}$ (a) and $P_{90}$ (b).

observed base rate for the $P_{00}$ and $P_{90}$ case, the $P_{99}$ case is positively biased. Compared to ERA5, cf. Figure S4, interannual

variability is increased. Furthermore, all trend lines point upward, most of them significantly. It is, however, noticeable - but possibly insignificant - that the warmest scenario SSP585 exhibits a weaker trend than the moderately warming SSP126. At first sight that may not seem to be in line with the narrative of warmer atmospheres generating heavier rainfall (e.g. Fowler et al., 2021), and we shall return to this point shortly. The results for the MPI-ESM1-2-HR model are less conclusive. Except for the extreme severity class $P_{99}$ the values are negatively biased against observations, and no scenario exhibits a significant

centennial trend. We have not been able to sort out any reason for this model behavior. Indeed, trend significance is more the exception than the rule, as of the 2400 cases only 610 are significant. Examples analogous to 7are Figs. S5-S8.



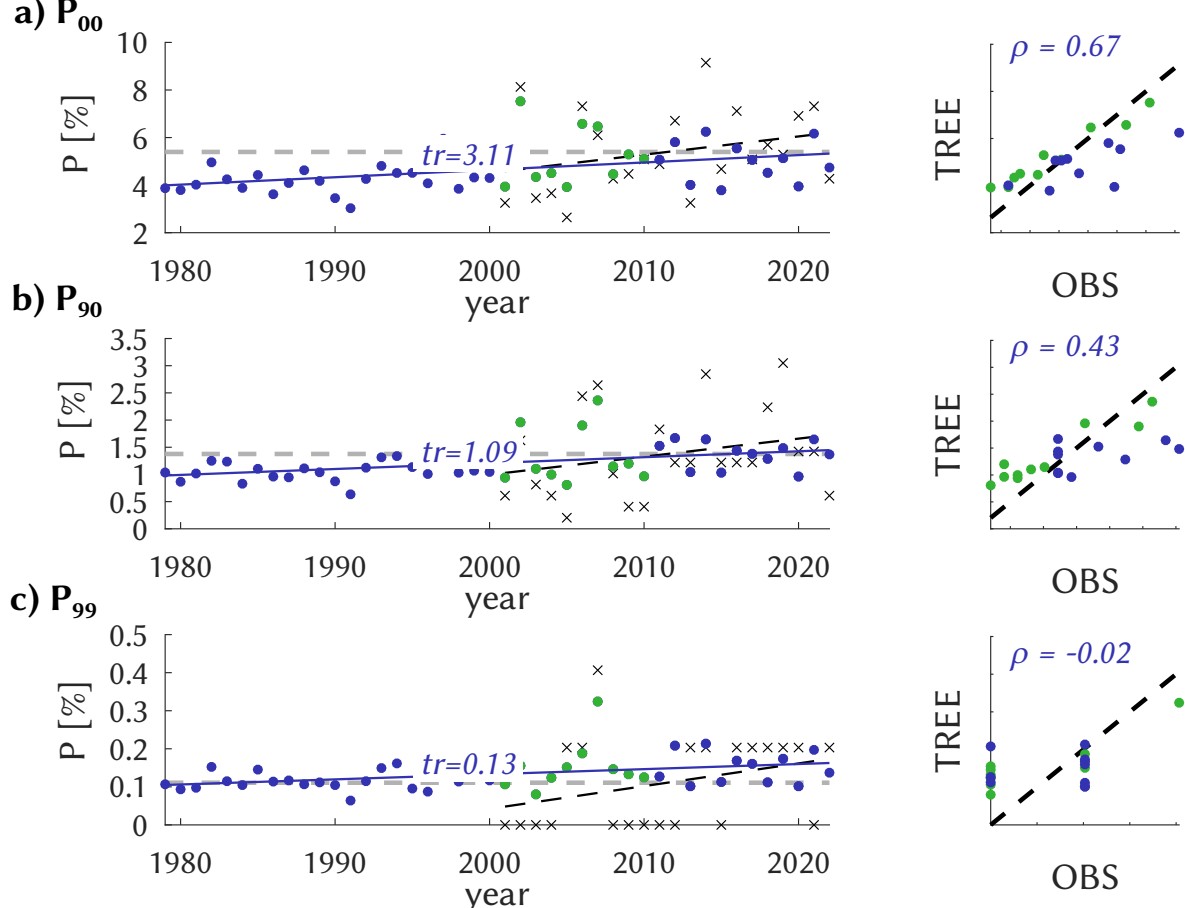

**Figure 6.** Left: Annual mean probabilities of class 1 occurrence for $P_{00}$ (a), $P_{90}$ (b), and $P_{99}$ (c) from observations (CatRaRE, black crosses) and from ERA5 using the TREE classifier as a time series (colored dots). Also shown is the centennial trend (solid if significant at 5 %). Right: The same as a scatter plot for the common period (2001-2022). The calibration period is indicated as green.

Returning to the CMCC-CM2-SR5-based trends and their consistency with the global warming narrative, we inspect annual values and trends of the corresponding atmospheric drivers, *cape*, *cin*, and *cp*. Indeed, Fig. 8 may seem to violate that narrative, as model-simulated convective precipitation (*cp*) increases in the moderate SSP126 scenario but decreases in the warmer SSP585. But one also notes that while for *cape* those two future trends (which are both significantly positive) are not substantially different, the *cin* trend is markedly stronger for SSP585 and appears to dominate the negative influence on *cp*. It means that rainfall *intensity* indeed increases due to the stronger *cape*, but at the same time that rainfall *frequency* decreases due to the stronger *cin*, a delicate balance that can lead to a positive or a negative effect on rainfall amount depending on GCM, region, etc.., in line with the law of Clausius-Clapeyron and the global warming narrative; see Chen et al. (2020), Myhre et al.




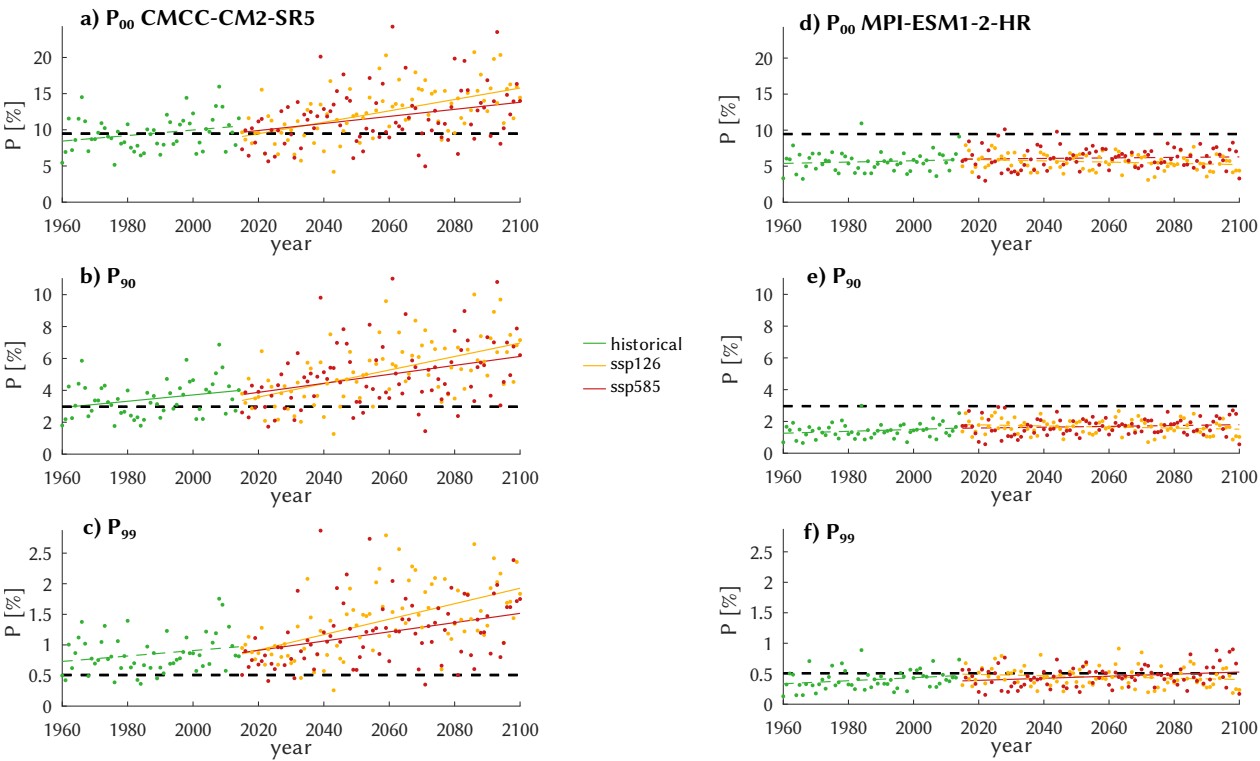

**Figure 7.** Simulated annual probabilities of the 3 severity classes $P_{00}$ (a,d), $P_{90}$ (b,e), and $P_{99}$ (c,f) occurring in the SW and corresponding centennial trends (solid if significant at 5 %), by applying the tree classifier to the scenarios HIST (green), SSP126 (orange), and SSP585 (red) of the two GCMs CMCC-CM2-SR5 (left) and MPI-ESM1-2-HR (right). Also shown is the observational base rate as obtained for the period 2001-2022 (black dashed).

(2019), and also section S3 of the SI. For the MPI-ESM1-2-HR model, corresponding trends are much weaker, and so is the negative influence of *cin*, leading to less divergence between the scenarios. More examples are Figs. S9-S12.

### 3.3 Distribution and ANOVA of the simulated trends

The distribution of the simulated trends is now investigated in the next figures. We start by showing in Fig. 9 the case of the classes with no regional focus, 0 (no events) and 1 (events anywhere), for the severity level $P_{90}$. Despite a considerable range of

180 uncertainty in all scenarios the majority 0-class decreases and the 1-class increases. The corresponding medians of the 0-class vary between -5%pt (for ERA5) and slightly below 0%pt (for SSP126). With a much smaller base rate the absolute class 1 trends are smaller as well, with a largest median from ERA5 just below 1%pt, a value that appears quite strong given a base rate of just 1.4 %, cf. Table 2; the GCM-based trends are smaller with medians varying near 0.2. The SSP585 trends for the



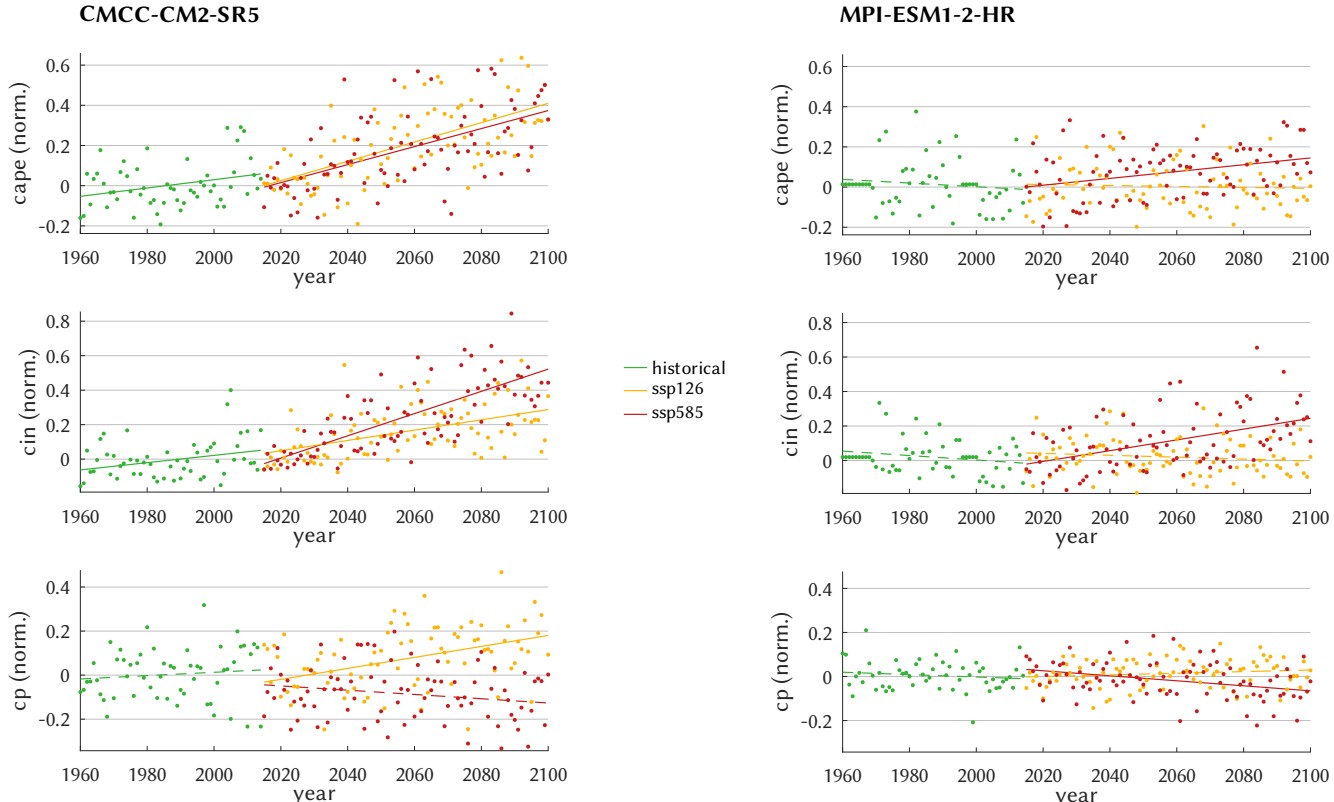

**Figure 8.** Similar to Fig. 7, for the 3 normalized drivers *cape*, *cin*, and *cp*.

0-class (1-class) are generally more negative (positive) than the SSP126 trends, as would be expected from greenhouse forcing.
The strong skewness for SSP126 may be related to the *cape* vs. *cin* antagonism that was potentially influential for the larger
CMCC-CM2-SR5 amplitudes. The results for $P_{00}$ are generally similar but with weaker amplitudes, cf. Fig. S13. Likewise, by
confining to significant trends the overall picture does not change much, merely the strength of the trends is stronger and more
in line with ERA5, cf. Fig. S14.

For the 4 quadrant regions, we depict trends relative to the respective base rate, to allow for a 'fair' regional comparison. The
190 overall message, shown in Fig. 10, is thus: trend uncertainty is large and the medians are positive throughout but below ERA5,
whose medians vary regionally about half of the base rate. And contrary to the example of Fig. 7, for all regions the SSP585
median is larger than that of SSP126 (consistent with Fig. 9), and HIST is between both. For $P_{00}$, see Fig. S15.

Each single trend is the result of a cascading set of model applications and assumptions, called 'factors', and their entirety
can to some degree be disentangled back into the single factors by employing the statistical machinery of ANOVA (=ANalysis
Of VAriance). For the GCM-based simulations (HIST, SSP126, SSP585), the obvious factors are severity (SEV), scenario
(SCN), climate model (GCM), classifier (DSC), and region (RGN); we do not consider factor interactions. One has to observe





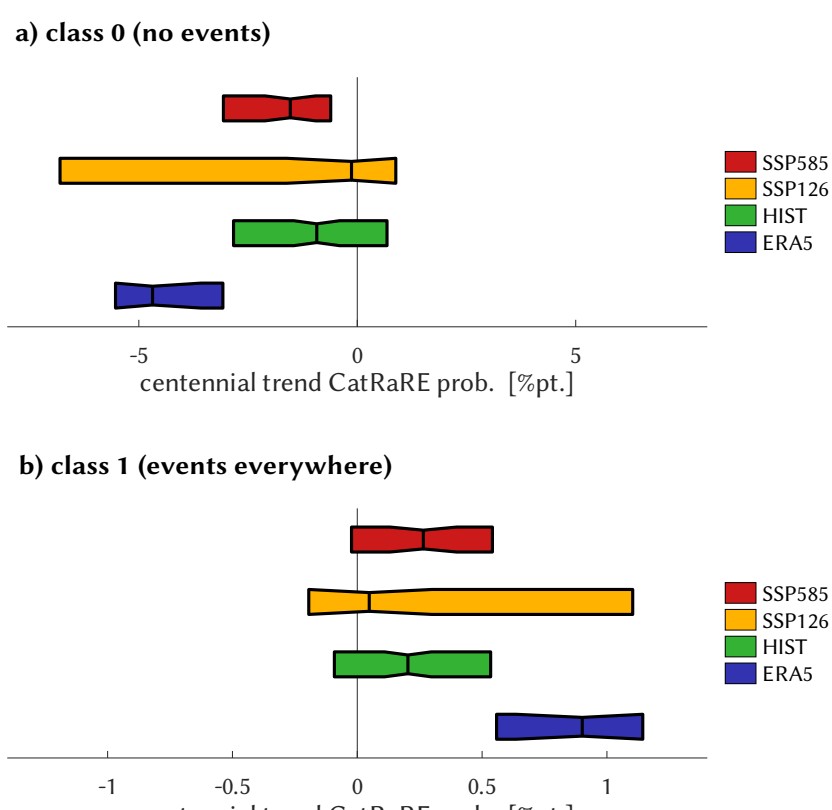

**Figure 9.** For ERA5 and the HIST, SSP126, and SSP585 scenarios, boxplots of the distribution of centennial trends of simulated annual CatRaRE-type probabilities of severity type $P_{90}$, for class 0 (a) and class 1 (b). Trend magnitude is given in percentage points. The width is determined by the inter-quartile-range and the vertical line depicts the median of the distribution. The different scaling of the x-axis is due to the much larger base rate of no events

that including all trends would put too much weight on the RGN class because of the mostly negative trends of the 0-class vs. the positive trends for the others. To have comparable RGN classes we confine the ANOVA to the four quadrant classes NW, NE, SE, SW. We are thus left with the abovementioned total of 2400 trend results (610 for significant trends). Fig. 11 shows the
outcome in terms of the F-ratios of the factors, a measure that expresses the between-sample variance (factor effect) relative to the within-sample variance (background noise). By far the greatest influence on the centennial trends comes from the GCM factor, followed by SCN; DSC and RGN are very small, which for DSC is astonishing given the great diversity of classifiers. It means that most of the uncertainty is contributed by the choice of GCM, even more than picking different emission scenarios. Fig. 12 shows the ANOVA based on significant trends only. Now the GCM F-ratio is largely reduced, and SCN remains the
sole dominating factor. In other words, the significant trends are driven mainly by the scenarios.





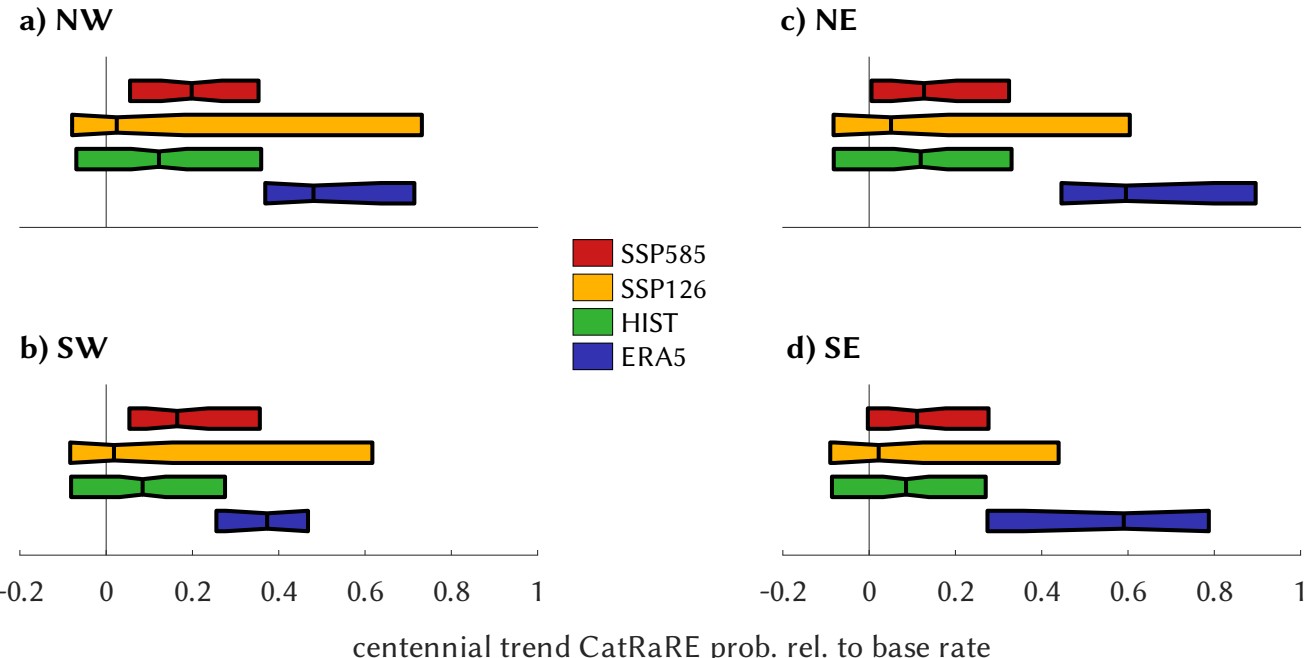

**Figure 10.** Similar to Fig. 9, for the 4 quadrant regions. Trend magnitude is given relative to the respective base rate.

## 4 Conclusions

We have investigated past and future trends in the occurrence of extreme rainfall events. Our modeling chain consists of a CMIP6 ensemble of simulated atmospheric convective environments that are fed into statistical classifiers to predict occurrence probabilities of CatRaRE-type events over the four quadrant regions of Germany. The classifiers are trained on corresponding

ERA5 fields, the learning target being the occurrence of CatRaRE events at a specified level of severity (all, upper 10 %, and upper 1 %). We are aware of, and have in the above analysis encountered, the trivial fact that increasing severity comes with a lower sample size and corresponding higher uncertainty. To counter this basic dilemma, we have employed an ensemble approach that tries to reflect as many aspects of the modeling chain as possible, and analyze the resulting distribution. We used as many of the current CMIP6 model data as we could retrieve (a number that is limited by the availability of upper

tropospheric fields). As classifiers we have employed a wide range of conventional and more novel ("deep") machine learning schemes.

Regarding the convective environments we have found growing levels of both antagonistic drivers *cape* and *cin*, cf. Fig. 8. This is in line with current knowledge and has been attributed to the fact that rising greenhouse gases are causing, at lower levels, an increase in specific humidity and thus more latent heating and buoyancy, but over land a decrease in relative humidity

and thus an elevated LFC, respectively, (cf. Eqs. 1 and 2, Chen et al. (2020)). The antagonism leads to higher uncertainty with respect to the future frequency of extreme convective events. But still, our analysis finds evidence that across Germany they





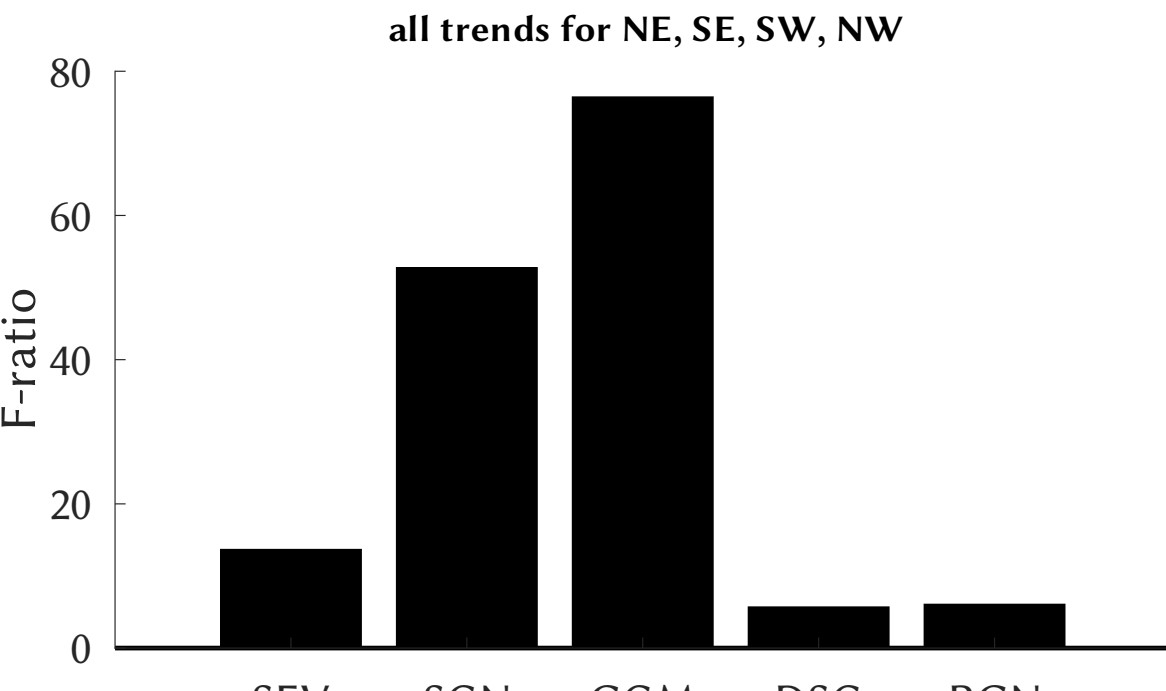

**Figure 11.** F-ratio of ANOVA of the centennial trends for any of the four quadrant regions over the factors severity (SEV), climate scenarios (SCN), climate model (GCM), classifier downscaling (DSC), and region (RGN). All factors (i.e. its F-ratio) are significant.

are occurring more often. This evidence, moreover, becomes clearer with growing levels of severity (Figs. 7, 9, 10), a fact that appears to be consistent with current observations, as exemplified by Fig. 1.

With respect to uncertainty in the centennial trends, the main factor appears to be the choice of climate model, and emis-
sion strength only second. This changes if insignificant trends are ignored, resulting in a largely reduced ensemble but with emissions now becoming the main factor for determining the trend. In both cases, the choice of classifier (downscaling) is less relevant, and so is the region considered. For downscaling this may be good news, as it means that very different approaches still arrive at similar results.

*Code availability.* The relevant code underlying this paper can be found at https://gitlab.dkrz.de/b324017/carlofff and is archived at Zenodo
(https://zenodo.org/records/16023089). Training and deployment of DL models is performed using the Caffe framework with its Octave interface (https://github.com/BVLC/caffe); using a corresponding container it can be run on various platforms.



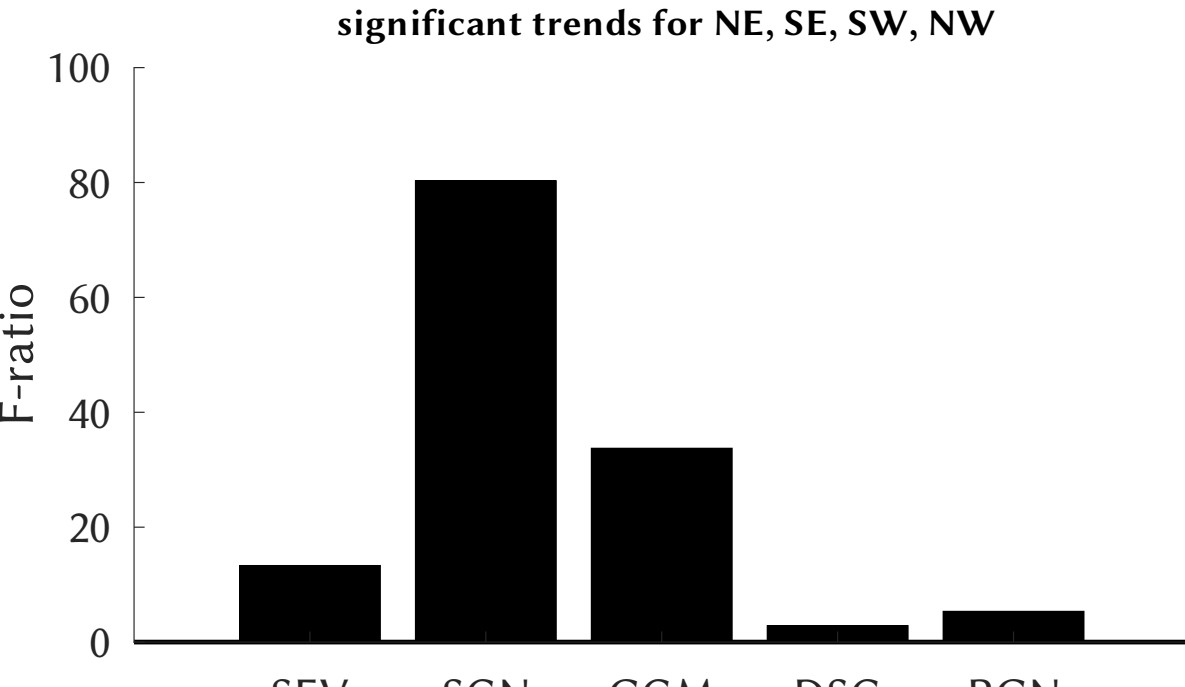

**Figure 12.** Like Fig. 11, confined to significant trends.

*Author contributions.* GB and MH envisaged the project, GB developed and conducted the modeling, GB and MH wrote the manuscript.

*Competing interests.* The corresponding author declares that there are no competing interests.

*Acknowledgements.* This research has been funded via the "ClimXtreme II" (sub-project CARLOFFF, grant no. 01LP2324B) by the German
Ministry of Research, Technology and Space (Bundesministerium für Forschung, Technologie und Raumfahrt, BMFTR) in its strategy
"Research for Sustainability" (FONA).



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
