# Peer review of "Present and future trends of extreme short-term rainfall events in Germany, by downscaling convective environments of ERA5 and a CMIP6 ensemble"

_EGUsphere, 2025_

## Referee Comment (RC1)

**Review for "Present and future trends of extreme short-term rainfall events in Germany, by downscaling convective environments of ERA5 and a CMIP6 ensemble"**

An article submitted by Gerd Bürger and Maik Heistermann

January 4, 2026

**1 General comments**

**1.1 Summary**

The manuscript *Present and future trends of extreme short-term rainfall events in Germany, by downscaling convective environments of ERA5 and a CMIP6 ensemble* submitted by Bürger and Heistermann proposes a procedure to obtain projections for (highly) aggregated information on extreme precipitation for regions in Germany from GCM projections. Basis is an extremness index $E_{T,A}$, which is a spatially and temporally aggregated characterisation of extreme precipitation events with a timestamp and location assigned. Events with $E_{T,A}$ over a certain threshold are used to classify every 6h interval into one of 6 categories, depending on the existence and location of the events. Additionally to these class labels, the authors use atmospheric information over a regular grid over Germany (CAPE, CIN, convective precipitation) obtained from ERA5 reanalysis to train classifiers which can map these 6h atmospheric information from ERA5 to one of the aforementioned 6 categories. Based on these classifiers, atmospheric information from GCMs can now be used to obtain future projections for the categories from GCM projections. Finally, they analyse the resulting projections of these categories for the 21st century for trends.

The study seems to be closely related to Bürger and Heistermann, NHESS, 2023 (henceforth BH), as also many descriptions of methods are defered to BH.

**1.2 Major comments**

I consider the underlying idea to obtain an aggregated and relevant quantity / index / statistic directly from GCM projections via a statistical model / machine learning approach as very useful! The alternative would be a numerically very costly modelling chain involving a dynamical downscaling of the variables needed (here precipitation) for the aggregated quantity (here the 6 categories) via several steps with an RCM and subsequently carry out the aggregation from the so downscaled precipitation. The latter

approach could not be easily carried out for a full century and the many GCMs considered in this manuscript.

As such, the paper addresses a relevant scientific questions within the scope of NHESS and present a novel approach. Manuscript and methods are up to international standards with some problems to be discussed in the next section. The scientific methods are on a high level but not sufficiently well outlined, at least not in a way that would enable me as a reviewer to understand what has been done, see next section.

The manuscript is refreshingly short as it defers many details to elsewhere. Which makes it difficult to review (and reproduce) as information needs to be found elsewhere. Furthermore, several steps are carried out in a non-standard or non clearly explaind way which allows to question the results, e.g. a linear gaussian model for quantiles instead of quantile regression (figure 1), a plethora of neural networks designed for image classification (without giving motivation why they are considered adequate here, table 3), particular implementation of logistic regression (see BH), modelling counts with linear gaussian models instead of Poisson- or similar regression, being not clear on how model climatologies have been obtained, using RPSS in a setting it has not been designed for (nominal categories).

I suggest to revise the manuscript based on the comments from the reviewers.

**2 Specific comments**

I can see their setting being similar to statistical downscaling. However, as the target is a categorical variable (1,R,0) which is neither available on the large scale, nor at a small scale and thus could also not be obtained directly from RCMs, I'd like to ask the authors if there is a better name / category for their concept. Something like "impact assessment" or "deriving convective precipitation related indices form GCMs"?

**2.1 Methods used and their description**

The authors defer for the description of methods to one of their previous articles (Bürger and Heistermann, NHESS, 2023, henceforth BH) in order to avoid repeating the description (even in the same journal). Although I support this idea in general, here it does not seem adequate as a) also BH is not very informative on several methods and b) details for the study at hand are missing.

In BH I found with respect to the description of "shallow" models: "As competitive benchmarks to DL models, we employ four shallow statistical models: lasso logistic regression (LASSO), random forests (TREE), a simple neural net with two hidden layers (NNET) and logistic regression based on non-linear least squares (NLS). All of these are applied with and without empirical orthogonal function (EOF) truncation, using North's 'rule of thumb' to find 33, 27 and 21 principal component predictors for cape, cp and tcw, respectively, as estimated from the calibration period; more details are listed in Table 1 and in the source code mentioned at the end." Unfortunately, from this information, I cannot evaluate how the authors actually used these models. With the current pressure

on the review system, it is difficult for a reviewer to go to the code and understand what has been done, particularly when the code is not written in the reviewers preferred language..

As an example, consider the logistic regression and LASSO logistic regression: In BH, the authors report only the LASSO cost function with the penalty term. Even if we assume that the reader knows the basic idea of logistic regreission (probabilities as target and logit-transformation), the following remains open: a) how do the authors deal with the 6 categories? The default logistic regression gives probabilities for two categories (event / no event), how do the authors get probabilities for 6 categories? Do they use multinomial logistic regression? Do they account for the fact that the 6 categories can not be ordered, i.e. they are nominal not ordinal? Or do they use binominal logistic regression for each of the six categories? The latter would not be self-consistent as probabilities for the 6 categories do not necessarily add up to 1. b) why does BH associates a "non-linear least squares" approach to logistic regression? To my knowledge, logistic regression is a special case of generalised linear models (GLMs) and can be solved in a likelihood framework with iteratively reweighted least-squares (IRLS), see e.g. Dobson and Barnett, "An Introduction to Generalized Linear Models" (2008). Is the approach given here equivalent? c) how does the predictor look like? Do cape, cin, cp for every grid point enter as terms in the predictor? Or do the authors use an EOF as they mentioned in BH? How is this EOF carried out? Are cape fields treated sepeartely from cin and cp as in BH? Or do all three fields enter the EOF simultaneously here? If analogously to BH, cin and cape are seperately. In the current manuscript, cp is used, in BH tcw. Should the reader now assume that cp is treated analogously to tcw? Is the truncation for cp the same as for tcw? Can terms in the predictor of the logistic regression interact? If not, the comparison of logistic regression to random forest would not be fair as for random forest factors interact there by default. To understand what how logistic regression has been used here, the reader need the model equation, e.g. $\text{logit}(p) \sim \text{cape} + \text{cin} + \text{cape} : \text{cin} + \ldots$.

Same for random forest: the reader needs to know what features can be used to define the split rules, are these cape and cin on the grid points or PCs from an EOF? How has that EOF been carried out? How many features can be used at each split? How is the data sampled for each tree? BH gives some information on that but does this transfer directly to the study at hand as we have a different set of predictors and predictant? I assume the reader expects an abstract (i.e. code independent) description of the details of the methods. Imagine that the reader is not familiar with matlab or octave but uses python or R instead. Thus a code independent description is needed. Saying that does not mean that it is not useful to publish the code!

I refrain from discussion the other methods in this way. Also because the neural network based methods can not be described in the way a logistic regression model can be decribed. I expect, however, that the reader needs an argument why a given method has been chosen for the canon of methods, e.g. giving typical usecases of the networks and a short explanation why this should be useful in the case at hand. I can imagine that image classification is similar to the case we have here with the three maps (cape,

cin, cp). Also color images have a grid structure with three values (RGB) at each grid points. So I see the similarity. Doing some research on LeNet-5, I find that this is used for very special images, namely of size 28x28, greyscale. Why is that considered here?

The ANOVA in Sec. 3.3 includes as factors GCM, region, scenario, classifier and severity. While I see GCM and classifier (and maybe region) as interesting factors to understand the uncertainty in the trend values, I do not understand why the authors use severiy and scenario as factors. I'd expect an ANOVA using GCM and classifier (and maybe region) sepearetly for various severity classes and scenarios to obtain an idea about the uncertainty for a given severity. Including severity as a factor seems like quantifying the uncertainty for trends of two different variables, such as for temperature and precipitation. Please correct me if I am wrong here.

**3 Technical corrections**

- Title: "... downscaling convective environments ..." are really the convective environments (caracterized by cape, cin, cp) transformed to a smaller scale? From my point of view, the study rather derives an impact related index directly from a GCM *without dynamical or statistical downscaling.*

- l. 14: 70B EUR → better 70 billion (most natural in running text) or €70bn or €70 bn ("bn" is preferred over a capital "B")

- l. 17: "evidence of this ...": "this" is related to what?

- l. 49 (and many other occasions): reading this out loud "99%-percentile" sounds strange as there is twice "percent". Suggestion: 0.99-quantile, 99%-quantile or 99. percentile. For higher quantiles use 0.999-quantile or 99.9%-quantile and 0.9999-quantile or 99.99%-quantile, respectively. Percentiles are associated with quantiles specified by integer percentages. The use of non-integer percentiles can be frequently seen but does not fit to the concept.

- l. 49: Are the trends obtained using quantile regression? If not I would be scepticle about the $p$-values. If a gaussian linear model is fit to quantiles, the assumptions might not fit and inference is not robust. Furthermore, the quantiles are themselves estimates with an uncertainty which should be accounted for. A gaussian linear approach still might give sensible estimates in some cases but quantile regression is definitively more appropriate here. As this is technical very simple, I suggest the authors use that to increase confidence in their study.

- l. 53: The authors argue that their trends are quite robust. What is meant here by robust? Removing some points lead to similar trends?

- l. 54-55: "... corresponding to the overall, 90% and 99% ..." the missing comma after overall seems very important

- l. 72: "As cape and cin are not provided by the CMIP6 models they had to be calculated from these profiles, following Bolton(1980)" $\rightarrow$ give more details on that in the supplementary material or an appendix.

- l. 79: why is the 10x10 grid used? This I did not understand. Maybe it has been explaind in BH? But the reader needs some information here.

- l. 82: "reference climate" this is not sufficient. Is the climatology obtained per day of the year? Per month? Or for every of the four time steps a day has (my favourite)? Or is it a mean over all time steps in the data set? Does every grid point has its own climatology (my favourite)? The text suggests that there is an average over all grid points. An equation would be unambiguous here.

- l. 91: What is meant by "probabilistically classified"? The classification does also need some more explanation. There are 6 nominal classes, i.e. they cannot be ordered. This has consequences on a) the logistic regression approach (multinomal regression) and b) the verification: RPS is not adequate as the classes (NE, NW, SE, SW) have no natural ordering but that is what the *ranked* probability score builds upon. $\rightarrow$ comment later.

- l. 102: I find the term "shallow" strange for a random forest and logistic regression as it puts these in a the framework of neural networks without any need.

- table 3: "Nonlinear least squares" as solver for the cost function related to logistic regression sounds strange, see my comments in 2.1. Why are there no number of parameters associated with the "shallow" models? If the authors put random forests and logistic regression in the frame of neural networks, why not giving the number of layers? LASSO, TREE and NLS have 1, NNET hast 2, right? Why has "logreg" 32x32 nodes? I do not find an explanation in BH, neither in the associated supplement.

- l. 106: *Exceedance counting* It is not clear what this chapter is good for.

- l. 107: "zero-order projection", as you extend the trend, I would call it a first-oder projection.

- figure 3 / l. 112: are these quantiles (percentiles) those resulting from the fit in figure 1? Or are these the overall temporal quantiles? What are the annual exceedance counts here? In l. 112 they are called "annual relative exceedance counts normalized by the overall exceedance count". This is not fully clear to me. I suggest an equation. This figure needs some more explanation. As these are counts, Poisson regression would be an appropriate trend model. Is this used here? Otherwise information on significance might be unreliable.

- l. 122: "cross-entropy" needs a reference, best for a similar case.

- l. 123: "Ranked Probability Skill Score (RPSS)" is the skill score based on the Ranked probability score, a score for probabilistic information for ordinal categories. Here we cannot order the categories and thus I doubt that the score is appropriate. Furthermore, the authors need to specify the reference prediction as needed for every skill score. Is it the climatology? I would be interested in seeing the reference for "RPSS is the skill score corresponding to the relative deviance of the predicted probabilities from the actual outcome".

- figure 4: RPS, see above. Why are only 12 methods shown? Table 3 gives 13 methods, however, it is not clear what is the difference between "Logistic regression (NLS)" and "Logreg (2023)". The latter is not showing up in figure 4. Why is there a range for RPSS? Is there a variation of model hyperparameters? Please explain this in more detail.

- l. 141: Please give the interpretation of the "averages of the resulting 6-hourly probabilities", e.g. expected number of events per 6h interval. Maybe it would be better to work with the sum over a summer period to give expected numbers for the summer?

- l. 152: "... the 20 entries of the table ..." which table?

- l. 160: if trends for probabilities are not obtained with logistic regression, I would be sceptical with the significance.

- l. 166: "... analogous to 7are Figs...." -> "... analogous to Fig. 7 are Figs...."

- figure 6: here are trends obtained for probabilities → use logistic regression. "Annual mean probabilities FOR class 1 ...". Explain the right column of the plot. How is OBS obtained?

- figure 8: Are these cape, cin and cp values averaged over the whole summers? Is this meaningful? It is short term strong cape which favours convection and cape is reduced through convective precipitation, so the mean does not need to be high but there can still be convective precipitation.

- l. 185: "The strong skewness for SSP126 may be related to the cape vs. cin antagonism" → Could that not be checked by looking at individual cases?

- l. 204: "Fig. 12 shows the ANOVA based on significant trends only" What is the argument for using significant trends only? The insignificant ones are either small and yield values close to 0, which would be ok for including them in the analysis, right? OR they are based on a small number of events (high severity class). This effect should be taken care of anyway by using only one fixed severity classes for an ANOVA.

- l. 214: "... aspects of the modeling chain ..." Yes, I agree. There are however more aspect to modelling chain presented here, which has not been considered, e.g. extremeness index, estimating quantiles or non-optimal regression approaches.

- l. 217: "... found growing levels of both antagonistic drivers cape and cin, ...". I do not see why the mean of these quantities over the whole summer should give arguments for that. I can imagine constant means over several summers but varying convective precipitation in each summer. As said earlier, we can have high cape which is eaten up by conv precipitation within a day.

- l. 227: "For downscaling this may be good news, as it means that very different approaches still arrive at similar results." Is this not in cotradiction with your RPSS figure? From my understanding, this means, that the variability due to different GCMs is just much larger than the differences bewtween classifiers. Do you really want to have this classification method related sentence as last sentence of your conclusion? Does that fit to your title? Should not a sentence about the trends at the end?